**Subject Category:**
Biology (whole organism)

behaviour

audio-visual communication, courtship dance, Estrildid finch, play, social development

**Author for correspondence:**
Masayo Soma
e-mail: masayo.soma@sci.hokudai.ac.jp

# Early-life lessons of the courtship dance in a dance-duetting songbird, the Java sparrow

Masayo Soma[1], Midori Iwama[2], Ryoko Nakajima[2] and Rika Endo[2]

[1]Behavioral Neurobiology Group, Department of Biology, Faculty of Science, and [2]Behavioral Neurobiology Group, Biosystems Science Course, The Graduate School of Life Science, Hokkaido University, Kita 10, Nishi 8, Kita-ku, Sapporo, Hokkaido, Japan

(iD) MS, 0000-0002-8596-1956

Vocal learners, such as songbirds, must practise singing in a developmentally sensitive period to master songs. Yet, knowledge remains limited about the development of visual displays in birds, even when courtship includes well-coordinated vocalizations (songs) and body motions. The Java sparrow (*Lonchura oryzivora*) is a species of songbird that exhibits a courtship duet dancing exchange between the sexes, with this behaviour driving mating success. In this study, juvenile male Java sparrows were observed in captivity, showing that they repeatedly practise the courtship dance in their early life. We called it 'practice', as juvenile birds frequently dance towards family members or other juveniles well before sexual maturation. Based on our observation that dance motor performance increased with age, we propose that the practice is needed for motor learning. In addition, it could also be important for establishing vocal-motional coordination or socialization. Older juveniles gradually became capable of singing and dancing simultaneously, and participated in duet dancing more often. We also found that repeated encounters with the same individual promote dance movement. Though our results do not show how much social experiences account for the development of dance communication, early-life dance practising might influence future reproductive success, like song practising does.

## 1. Introduction

Avian courtship displays are often expressed as multimodal signals that are composed of complex vocalizations and ritualized body motions (often referred to as courtship/mating dance) [1–6], but

their developmental mechanisms have been identified mostly for vocalizations. Especially in passerine species, song development and its social process have been well determined. For instance, young oscine males acquire songs by memorizing and practising to imitate the phonology of adult song models through sensory and motor learning periods [7], with both auditory stimuli and social interactions playing an important role [8,9]. By contrast, it is not known whether oscine males follow similar behavioural development to master dance displays, or if they are able to perform them without 'practising' once sexually mature, cf. [10].

Although there is virtually no systematic study of the ontogeny of dance-like displays in birds, several studies have mentioned how young birds perform them even before breeding age (e.g. crane [11]; riflebird [12]; manakin [13]; bowerbird [14]). These reports, which include various taxonomic groups of birds, suggest that 'practising' dance might be more widespread than we assumed.

Why some birds need to practise dance well before reaching breeding age is a mystery. One possibility is that young songbirds might perform dances during the song-learning period to rehearse coordinated singing and dancing. In addition, it is also possible that young birds need to 'practise' dance for motor training, just as they do for the acquisition of songs [7–9,15,16]. These hypotheses highlight the similarity with animal play, in that dance practices are shown by young individuals and potentially contribute towards improving physical skills or the acquisition of social bonding [11,17,18].

Questions on the adaptive significance of dance practice become more intriguing when considering song/dance duetting species. For territorial defence, mate guarding, or courting, some songbird species engage in duetting, in which a male and a female show well-coordinated singing behaviour [19,20]. In comparison, the pairing partners of non-songbird species, like cranes, grebes and boobies, perform duet dances [21–23]. Moreover, some species perform multimodal duetting [3,24]. These duets are often characterized by the precise temporal coordination of behaviours of paired birds, and apparently need some form of learning and practising [25,26]. Although empirical evidence showing how duetting behaviour longitudinally changes over time is scarce, some studies support the concept that experience shapes duets. For instance, well-synchronized duets by magpie-larks (*Grallina cyanoleuca*) are attributed to long-term pairs [24]. In comparison, the juveniles of canebrake wrens (*Cantorchilus zeledoni*) practise song duets by participating in the duets of adults and gradually improving their temporal coordination [27]. In addition, similar vocal interactions have been reported between mothers and daughters of a gibbon, a duetting ape [28].

The Java sparrow (*Lonchura oryzivora*; Passeriformes: Estrildidae) is a songbird species that is characterized by duet dancing [29]. Java sparrows of both sexes often show courtship dancing in a mutually interactive way, wherein only the males sing [30,31]. In our previous study, duet dancing was frequently observed before successful mating [29]. Specifically, when they court prospective mates that are nearby (often on the same perch), males exhibit a series of repeated hopping and bill-wiping behaviours before and during singing. Subsequently, females often respond by exhibiting the same behaviours, followed by the copulation solicitation display and male mounting. Because the two main behavioural elements (i.e. hopping and bill-wiping) are highly stereotyped and are shared among the conspecifics of both sexes, these behaviours define courtship dance [29]. This dance (hopping and bill-wiping) is occasionally exhibited by juvenile Java sparrows without mounting or copulation (figure 1; electronic supplementary material, movies S1 and S2). As an initial step to elucidating the function of such dance practising in juvenile birds, in the current study, we traced the development of courtship behaviour in captive Java sparrow males. In particular, we aimed to reveal: (i) age-related changes of dance practising in relation to the song learning period, (ii) the social factors that promote dance practising, and (iii) age-related changes in the occurrence of duet dancing.

## 2. Material and methods

We obtained subject birds from 14 cage breeding pairs. A total of 28 laboratory-bred individuals (13 males and 15 females) from 14 different broods were reared in 13 broods by the cross-fostering of eggs. Cross-fostering was used to separate the effects of shared social environment and kinship on song development and, potentially, dance development. These birds were reared in family cages with their foster parents and one or two siblings (i.e. brood size = two or three), which were placed in one bird room. Nine males were used as focal subjects and were observed regularly from fledging to sexual maturity (figure 2*a*), while the other males were mainly used as pairing partners for the observation of focal males. Four focal males in a pilot study were observed once every 4 days from 24–28 dph (days post hatch) to 170–191 dph. All other focal males were observed once a week

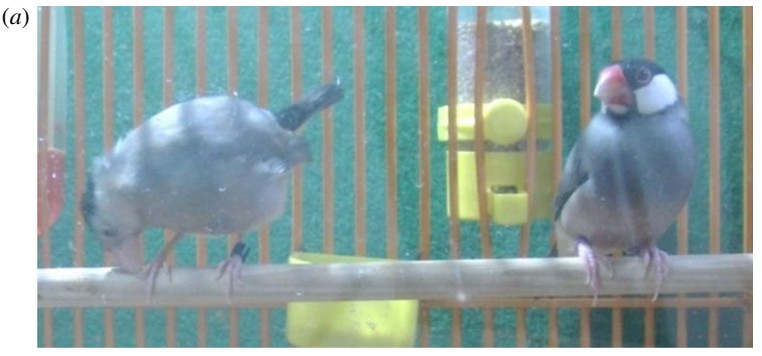

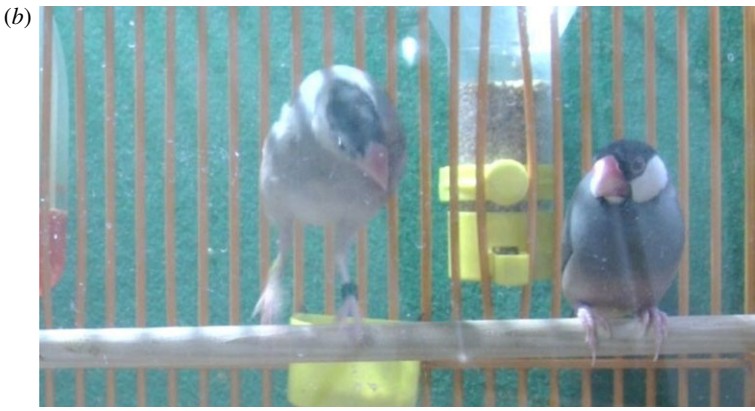

**Figure 1.** Dance practising. A juvenile male Java sparrow (106 days post hatch, left) exhibiting courtship dance behaviour beside an adult female, with repeated bill-wiping (*a*) and hopping (*b*). Also see electronic supplementary material, movies S1 and S2 and [27].

from around 40 to 180 dph. After confirming that these differences in sampling did not bias the statistical outcomes, we pooled the data from 40 to 180 dph for all focal males for the statistical analyses.

Subject birds were observed consecutively for 5 h using a video camera (Q3HD, ZOOM; iVIS HS R31, canon; GC-PX1, JVC) when they were either in natal family cages with their parents and siblings (family condition) or in experimental cages paired with a partner bird (paired condition), placed in a sound-attenuated chamber. When the subject birds were young, we used the family condition, because, based on our experience, temporary isolation of fledglings from their parents tends to solicit major parental aggression toward the fledglings when they returned. The aforementioned four birds used in a pilot study continued to be observed under family condition as well until maturity (after 180 dph), while the others switched from family condition to paired condition at around $64.3 \pm 3.5$ dph. In the paired condition, each focal male was paired with a male or female peer, raised in a different family. All family cages in the bird room were visually, but not acoustically, isolated from others using partitions; thus, the pairing partners and focal males met for the first time during this experiment. Because pairing partners that were sometimes recorded for other experiments were unavailable, it was not feasible to design a balanced combination of focal males and pairing partners. Overall, each focal male was paired with each partner 1–19 times (mean $\pm$ s.e. $= 3.8 \pm 0.6$ times) throughout the study. Therefore, the effects of repeated pairings with the same partner and the sex of the partner were statistically considered.

## 2.1. Dance occurrence analyses

From the recorded video of the 5 h observation period, we measured the occurrence of the courtship dance and whether it was accompanied by singing. One dance bout was defined as a behavioural sequence that was composed of at least three bill-wiping motions and one hop that occurred at short-intervals (less than 5 s) (figure 1).

First, we examined age-related changes in the dance frequency of focal males. We adopted a generalized linear mixed model (GLMM) [32] with binomial distribution to analyse the occurrence of dance (0–1 data) in the 5 h observation period. If one or more dance bouts were observed, it was scored as presence (1), and no dance as absence (0). In this statistical analysis, bird identity was

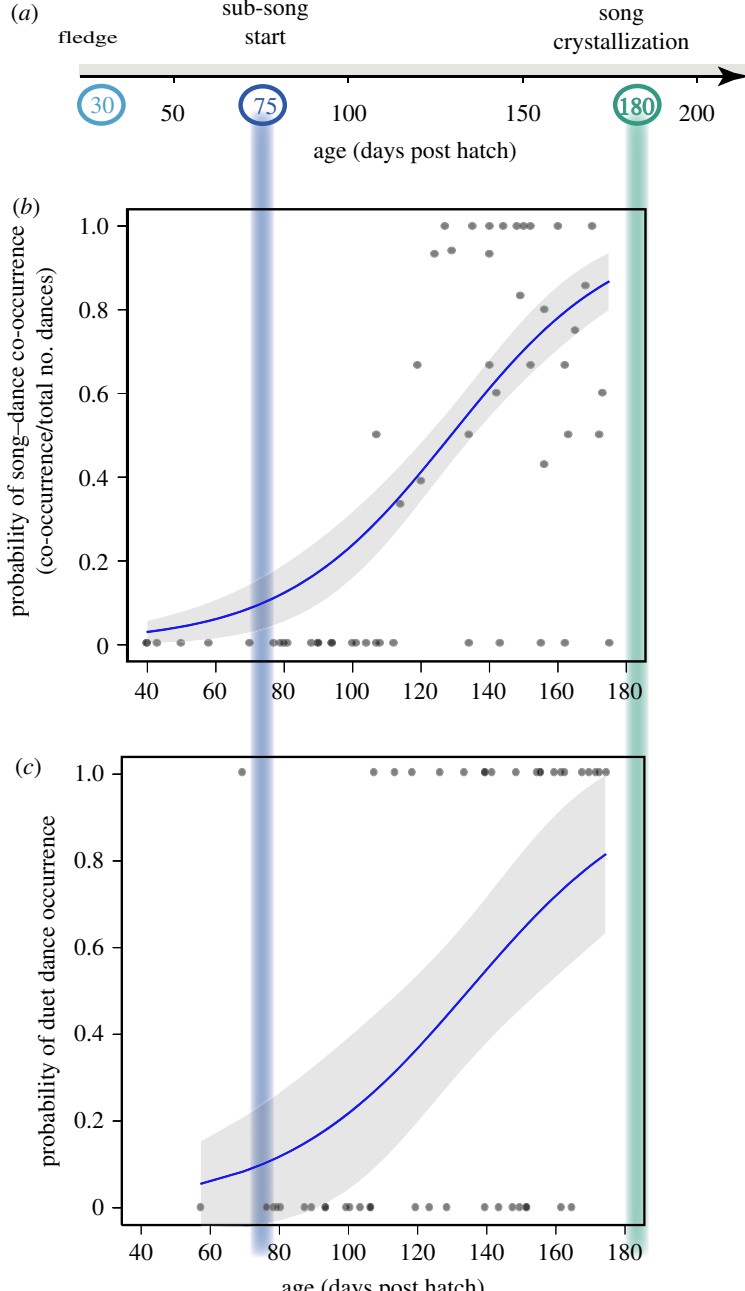

**Figure 2.** Development of song by Java sparrows (*a*). Age-related changes in dance–song co-occurrence (*b*), and the occurrence of duet dance (*c*). The blue lines were estimated from the corresponding GLM with binomial distribution (logit link), and shown with 95% confidence intervals (grey bands).

entered as a random effect to address the non-independence of the data taken from the same individuals, while age (dph) and experimental condition (family or paired) were entered as fixed effects. In order to deal with the confounding effects (i.e. lack of early age data for paired condition), we repeated the same analysis using subset data, which were taken from the four pilot study birds between 60–180 dph, as they were observed under both paired and family conditions during the period. In addition, to examine age-related changes in dance–song co-occurrence, we analysed the proportion of dance bouts that accompanied songs (total no. dance bouts with songs/[total no. dance bouts with songs + total no. dance bouts without songs]), using a binomial GLMM, where the same set of random and fixed effects as the previous analysis was entered.

Second, to explore social factors related to the development of dance, we extracted the data for the paired condition. We tested whether the occurrence of dance (0–1 data) differed depending on the sex

of the pairing partner or repeated encounter with the same partner using a GLMM with binomial distribution with random effects of partner identity nested within bird identity. Repeated encounter was scored as ordinal position that he repeatedly met a specific partner (e.g. 1: first time pairing, 2: second pairing, and so on).

Finally, we examined age-related changes in the occurrence of duet dancing. We were particularly interested in whether dancing by focal males constituted duets or not. Thus, we extracted paired condition data with focal male dancing and tested whether duet occurrence was influenced by the age of the focal male, the number of dance bouts that the focal male exhibited in a 5 h observation period, dance–song co-occurrence, partner sex, and the order of encounters with the same partner. We used a GLMM with binomial distribution with the random effect of bird identity. Given that this model involved a relatively large number of explanatory variables, we calculated variance inflation factor (VIF) [33], and found that there was no sign of serious multicollinearity (VIFs < 5). We also confirmed that simpler models that involve age and one explanatory variable only returned consistent results with that of the full model.

## 2.2. Dance performance analyses

In order to examine whether dance performance improves with age, we focused on three subject birds that showed dance regularly throughout the observation period. We measured the duration of each dance bout, relying on the same definition of dance bout as the above analyses. We also estimated the dance speed by measuring the intervals between successive hops or bill-wipes included in dance bouts (referred to as hop interval and bill-wipe interval, respectively; figure 3a,b). We adopted linear mixed-effect (LME) models [34], where we tested the effect of age on dance duration, and hop or bill-wipe interval, with the random effect of bird identity.

All of the statistical analyses were conducted using R v. 3.5.2 (R Core Team 2018). We used package 'lme4' for GLMM, 'nlme' for LME, and 'car' for calculating VIF. Overdispersion was not detected in any GLMM, for which we used 'overdisp' function in package 'sjstats' [35]. Statistical significance of each effect in LME and GLMM is based on full model with an alpha level of 0.05.

# 3. Results

## 3.1. Age-related changes in dance in relation to song

The earliest dance was observed at 40 dph (days post hatch), which was soon after fledging and much earlier than the time at which Java sparrows usually start to sing sub-songs (figure 2a,b). As the birds matured, they were more likely to dance ($p < 0.05$, GLMM; table 1a,b), and later exhibited dance and song simultaneously (less than 0.001, GLMM; table 1c, figure 2b). The earliest observation of dance–song co-occurrence was 107 dph (figure 2b). Previously, they danced without songs (electronic supplementary material, movie S1). In addition, we found that the paired condition elicited more dance than family condition (table 1a,b), though we lacked data on very young birds in the paired condition (see Material and methods).

## 3.2. Social factors that affect dance practice

When focusing on the paired condition, the birds tended to show dance when they were paired with the same partner repeatedly ($p = 0.002$, GLMM; table 2), but partner sex did not have a statistically significant influence on this parameter (table 2).

## 3.3. Dance development and duet

When we focus on when the focal males danced in the paired condition, duet dance was more likely to occur when they were older ($p = 0.015$, GLMM; table 3, figure 2c). In addition, repeated encounters with the same partner suppressed the duet; however, all other explanatory factors considered did not have statistically significant influences (table 3).

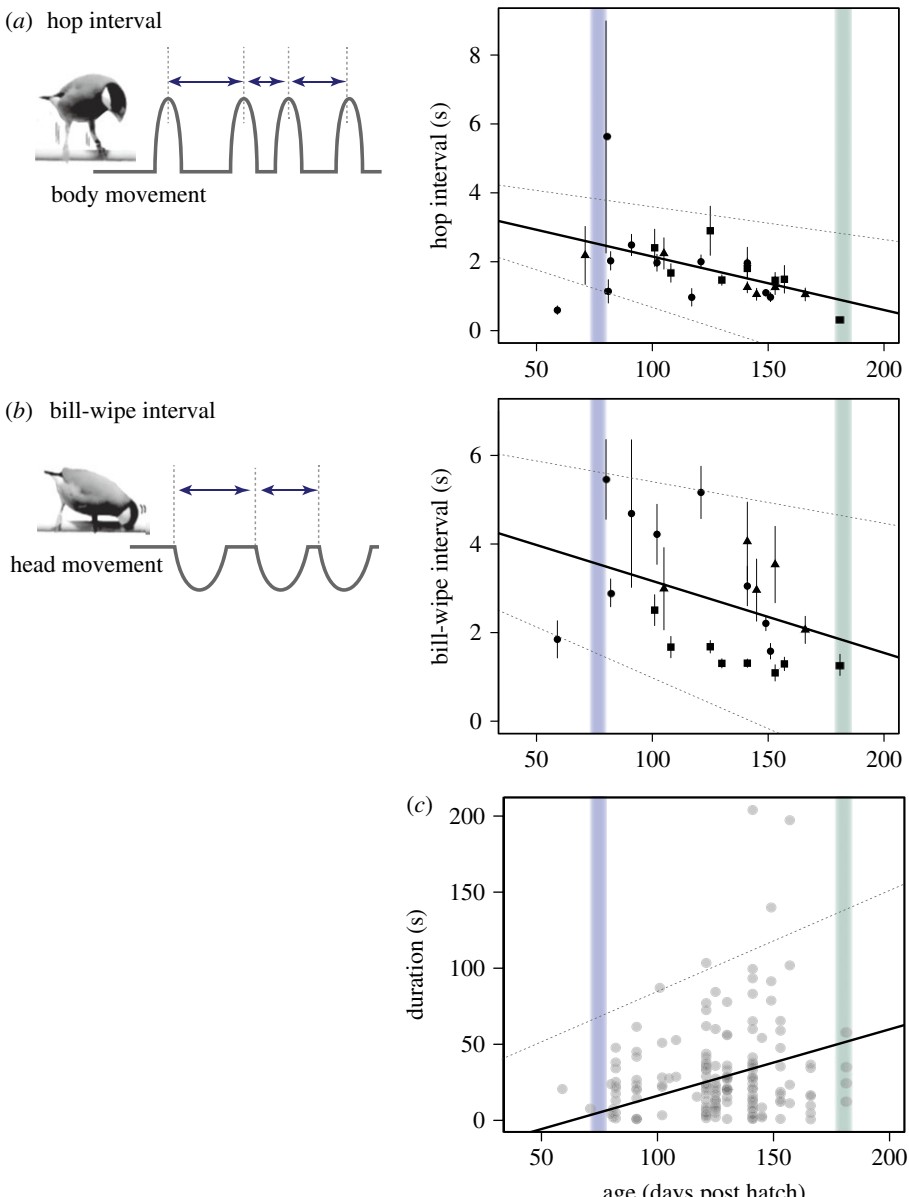

**Figure 3.** Age-related changes in dance speed, measured based on hop interval (*a*) and bill-wipe interval (*b*), and dance bout duration (*c*). Schematic images (*a,b*) show how the intervals were measured. Hop and bill-wipe intervals are plotted as mean $\pm$ s.e., using different point symbols for different birds (*a,b*). Estimated relationships are drawn with solid lines, with 95% confidence intervals (dashed lines). Blue and green vertical lines give an indication of the song development period of the Java sparrow (as in figure 2).

## 3.4. Age-related changes in dance performance

The focal three males showed on average $7.4 \pm 1.5$ dance bouts in the 5 h observation period, which corresponded to $128.8 \pm 43.6$ s (range $= 7.7{-}823.4$ s/5 h) dancing in total. As the birds matured, their dance speed got faster with shorter intervals between successive hops and bill-wipes ($p < 0.001$, LME, table 4*a,b*, figure 3*a,b*), and their dance duration increased ($p < 0.001$, LME, table 4*c*, figure 3*c*).

# 4. Discussion

Our results suggest that Java sparrow males invest considerable time in practising courtship dances in their early life. We call it 'practice', because juvenile birds dance frequently towards their family members or juvenile individuals regardless of sex before sexual maturation, and this behaviour does not lead to copulation. Although such behaviour in immature birds has been described in some

**Table 1.** Results of GLMMs testing the effects of age and experimental condition on dance occurrence (a,b), and dance–song co-occurrence (c). Positive values for the coefficient of condition indicate increases in paired condition, as shown in body type. Subset data (b) were taken from the four pilot study birds between 60–180 dph under both paired and family conditions.

| (a) dance occurrence | | (n = 212 observations, 9 birds) | | |
|---|---|---|---|---|
| fixed effect | coefficient | s.e. | z | p |
| intercept | −3.586 | 0.786 | −4.56 | <0.001 |
| age (dph) | 0.012 | 0.005 | 2.54 | 0.011 |
| condition (pair or family) | 1.155 | 0.460 | 2.51 | 0.012 |
| random effect | variance | s.d. | | |
| bird ID | 1.354 | 1.163 | | |
| (b) dance occurrence (subset data) | | (n = 110 observations, 4 birds) | | |
| fixed effect | coefficient | s.e. | z | p |
| intercept | −3.890 | 1.048 | −3.71 | <0.001 |
| age (dph) | 0.014 | 0.006 | 2.10 | 0.036 |
| condition (pair or family) | 1.606 | 0.603 | 2.67 | 0.008 |
| random effect | variance | s.d. | | |
| bird ID | 0.159 | 0.340 | | |
| (c) dance–song co-occurrence | | (n = 212 observations, 9 birds) | | |
| fixed effect | coefficient | s.e. | z | p |
| intercept | −7.336 | 1.513 | −4.85 | <0.001 |
| age (dph) | 0.048 | 0.008 | 6.00 | <0.001 |
| condition (pair or family) | 1.252 | 0.983 | 1.27 | 0.203 |
| random effect | variance | s.d. | | |
| bird ID | 1.400 | 1.183 | | |

**Table 2.** Result of GLMM on the effects of repeated encounters with the same partner (order) and partner sex on dance occurrence.

| | | (n = 136 observations, 9 birds) | | |
|---|---|---|---|---|
| fixed effect | coefficient | s.e. | z | p |
| intercept | −2.935 | 1.097 | −2.68 | 0.007 |
| order (with the same partner) | 0.294 | 0.096 | 3.09 | 0.002 |
| partner sex | 1.119 | 1.113 | 1.01 | 0.314 |
| random effect | variance | s.d. | | |
| partner ID | 2.651 | 1.628 | | |
| bird ID | 4.149 | 2.037 | | |

species [11–14], this study is the first to demonstrate quantitatively the longitudinal development of the visual courtship display in relation to singing, and to explore the social roles of this behaviour. Java sparrows start dancing before singing sub-songs, and then gradually incorporate singing into dancing and exhibit duet dancing more frequently (figure 2). They also improve dance motor performance during this period (figure 3). Thus, the adult-like courtship dance is acquired through practice.

Both dance and song were expressed in early life (figure 1), but follow rather different developmental trajectories. Soon after fledging, Java sparrows performed dancing that already involved essential behavioural elements (i.e. hopping and bill-wiping). Thus, dance motions are intrinsic and the process of dance development might not involve the sensory-learning (observational learning) period that is required for song acquisition. However, it is likely that young birds must dance for motor training (figure 3), or for coordinating a simultaneous expression of singing and dancing, cf. [10,36] (figure 2b).

**Table 3.** Result of GLMM testing multiple factors on the occurrence of duet dance.

| | | ($n = 46$ observations, 6 birds[a]) | | |
|---|---|---|---|---|
| fixed effect | coefficient | s.e. | z | p |
| intercept | $-7.714$ | 3.196 | $-2.41$ | 0.016 |
| number of dance bouts | $-0.051$ | 0.088 | $-0.58$ | 0.562 |
| age (dph) | 0.078 | 0.032 | 2.43 | 0.015 |
| dance – song co-occurrence | 1.286 | 1.647 | 0.78 | 0.435 |
| order (with the same partner) | $-0.592$ | 0.280 | $-2.11$ | 0.035 |
| random effect | variance | s.d. | | |
| bird ID | 0 | 0 | | |

[a]Three subject birds did not dance in the paired condition.

**Table 4.** Results of LME analysis on age-related changes in hop interval (*a*), bill-wipe interval (*b*), and dance bout duration (*c*).

| (*a*) hop interval (s) | | | | ($n = 2178$, 3 birds) |
|---|---|---|---|---|
| fixed effect | coefficient | s.e. | t | p |
| intercept | 3.704 | 0.448 | 8.275 | $<0.001$ |
| age (dph) | $-0.016$ | 0.003 | 4.822 | $<0.001$ |
| random effect | variance | s.d. | | |
| bird ID | 0.018 | 0.135 | | |
| (*b*) bill-wipe interval (s) | | | | ($n = 2277$, 3 birds) |
| fixed effect | coefficient | s.e. | t | p |
| intercept | 4.789 | 0.764 | 6.268 | $<0.001$ |
| age (dph) | $-0.016$ | 0.003 | 4.687 | $<0.001$ |
| random effect | variance | s.d. | | |
| bird ID | 1.030 | 1.015 | | |
| (*c*) dance bout duration (s) | | | | ($n = 166$, 3 birds) |
| fixed effect | coefficient | s.e. | t | p |
| intercept | $-27.578$ | 0.395 | 1.585 | 0.115 |
| age (dph) | 0.437 | 0.003 | 3.593 | $<0.001$ |
| random effect | variance | s.d. | | |
| bird ID | 149.50 | 12.23 | | |

Even so, we still cannot answer the question of why they start dancing approximately one month before they start singing, and why they need a longer practising time for dancing than for singing. Furthermore, it also remains unknown why Java sparrows hardly practise dancing without conspecific individuals while they frequently practise singing when alone (M.I. and M.S. 2014, personal observation).

The courtship dance that is mutually exchanged between males and females (duet dance) is a crucial predictor of mating success in the subject species [29]; thus, we expect that practising dance by juveniles involves some social process. Supporting this hypothesis, we observed that juvenile birds tended to dance with individuals that they repeatedly encountered, which indirectly indicated that social facilitation influences dance. In some duet-singing species, like songbirds and gibbons, young individuals develop/learn the rules of duets through vocal interactions with adults [27,28], which might explain the adaptive function of practising dance by juvenile Java sparrows. However, we have yet to determine whether duet dancing by the Java sparrow is governed by precise temporal rules like vocal duets. It seems that the duet dancing of Java sparrows is improvised rather than stereotyped (electronic supplementary material, movies S1 and S2); thus, juveniles might gain interactive experience, which improves their ability to dance with potential mates when mature. In addition, it is

possible that premature dance has communicative functions, and contributes to the formation of social or pair bonding. To determine what is actually gained through the social interactions of dancing in juveniles, we would need to design a study that controls for social experience.

This study highlights the analogies between juvenile dance in the Java sparrow and social play in mammals [11,17,18]. Practising dance is costly in terms of energy and time, but might contribute towards mastering adult-like dance displays or socialization, which might yield some reproductive pay-offs in the future. To understand the outcomes of 'practising', future studies should investigate the relationship between practising effort and reproductive performance.

Ethics. This study was conducted with approval from the Institutional Animal Care and Use Committee of the National University Corporation at Hokkaido University (no. 11-0028) in accordance with Hokkaido University Regulations of Animal Experimentation. During the study, stress was minimized and all birds were cared for and treated appropriately in accordance with the Guidelines for Proper Conduct of Animal Experiments from the Science Council of Japan and the Guidelines for Ethological Studies from the Japan Ethological Society.

Data accessibility. The datasets supporting this article are stored in the Dryad Digital Repository: https://doi.org/10.5061/dryad.g174p30 [37].

Authors' contributions. M.S. and M.I. designed research; R.E., R.N. and M.I. collected the data; M.S. analysed the data and wrote the paper.

Competing interests. We have no competing interests.

Funding. This study was supported by Japan Society for the Promotion of Science Grants-in-Aid for Young Scientists (nos. 23680027, 16H06177) received by M.S. All authors gave final approval for publication.

Acknowledgements. We thank Mari Shibata for the Java sparrow illustrations and the three reviewers for giving us a number of constructive comments.

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
