## [Reviewer comments · Royal Society Open Science]

Review History

RSOS-180980.R0 (Original submission)

Review form: Reviewer 1 (William Searcy)

Is the manuscript scientifically sound in its present form?

Yes

Are the interpretations and conclusions justified by the results?

No

Is the language acceptable?

Yes

Is it clear how to access all supporting data?

Yes

Do you have any ethical concerns with this paper?

No

Have you any concerns about statistical analyses in this paper?

No

Recommendation?

Reject

Comments to the Author(s)

The manuscript reports on an observational study of dance development in the Java sparrow. Dance development in songbirds has been too little studied, and is potentially important especially because of possible comparisons with song development. A limitation of the present study is that only the simplest measurements of dance were made: whether dancing occurs, whether dancing is paired with song, and whether dancing occurs in a duet with a partner. Without more sophisticated measures of dance performance, the study cannot really show that performance improves through the period of practice. By contrast, studies of song duets have tested whether temporal coordination and duet code adherence improve over time. With dance duets, some analog of temporal coordination presumably could be measured, but that was not attempted here. A further step would be to study dance development experimentally, in the way that has been with song development starting with the classic studies of Thorpe and Marler. For example, one could isolate young birds and study dance development in the absence of external models – but one would still have to have more sophisticated measures of the dance to decide whether the form of the dance is altered by isolation.

The authors do provide convincing evidence of certain developmental trends: the frequency of dancing goes up with age, co-occurrence of dancing with song increases with age, and duet dancing increases with age. But given the limitations of the dance measurements that are made, not a lot can be concluded from these results. There is very little indication on whether dancing skill increases with age, on whether practice is necessary to dance properly, or on whether young birds learn anything from observing others dance. The authors have no evidence on whether dance in young birds has a communicative function or is necessary for coordinated singing and dancing (hypotheses mentioned in their introduction). The developmental trends that are observed could be due simply to changes in motivation to dance combined with the known developmental patterns of song.

Lines 15-16: I don't see what in the results suggests that early social experience is essential for the development of courtship. The authors would have to make an explicit argument on this point if they want to include this claim.

Lines 36-40: these statements are hypotheses, not predictions. I'm not sure why these hypotheses are mentioned here since they are not tested in the paper.

Lines 58-59: the previous study does not show that duet dancing is crucial to mating success in the sense of causing higher mating success. The authors might say that duet dancing is associated with mating success in that study, but the results of that study do not show that duet dancing is causal.

Line 91: to say that birds were "audibly isolated" means that one could hear that they were isolated. A better wording would be to say is that the birds were not "acoustically isolated."

Line 143: Can you give an estimate of time investment in dance practice? It does seem that you have the data to do so.

W. A. Searcy
University of Miami

Review form: Reviewer 2 (Alex Jordan)

Is the manuscript scientifically sound in its present form?

Yes

Are the interpretations and conclusions justified by the results?

No

Is the language acceptable?

Yes

Is it clear how to access all supporting data?

No

Do you have any ethical concerns with this paper?

No

Have you any concerns about statistical analyses in this paper?

No

Recommendation?

Accept with minor revision (please list in comments)

Comments to the Author(s)

This is a well written piece that touches on numerous aspects of important theory with respect to the development and expression of courtship behaviour in birds. I find the introduction to be very compelling, setting up the study nicely. The authors ask whether the 'dancing' aspects of courtship develop over time, in a similar fashion to courtship song, showing convincingly that it does. I am not a bird person so am not certain of their claims this is the first time this has been shown, but nevertheless the data support the claim. Secondly the authors ask whether social conditions affect the development of dance and the co-occurrence of song and dance. Here is where the study and interpretation get a bit more tricky. First, the authors change the housing conditions of juveniles for brief periods during development - keeping them with the family unit (which is confusingly not defined in the main text, does it contain other juvenile birds?) or the paired condition in which birds are kept with other juveniles (male or female). They ask whether the onset of dance (confusingly called 'occurrence' but should probably be changed to 'onset'), the rate of dance ('number of dance bouts'), and the co-occurrence of song and dance are affected by these social contexts. From these treatments and measurements, my take away message is that the number of dances increases in the paired condition, which may be a consequence of greater opportunities or motivation when meeting a juvenile (because whether this is the only time is see a juvenile or not I cannot tell as I don't understand whether the family condition also contains juveniles), or may be a function of the familiarity of the juvenile partner as the authors claim - the frequency of dancing may be higher when the focal bird does not know its partner, at least initially, but then becomes familiar and increases the frequency of bouts with better known birds. The co-occurrence of dance and song increases with age, but not with social conditions.

While I have some reservations about how well the methodology addresses some of the points raised in the introduction, I nevertheless think the paper warrants publication after some small changes to better explain the differences in conditions and opportunity for courtship interactions in the treatments. The main point I would like to see the authors address is how much evidence they have that there is any aspect of social learning going on that cannot be explained by developmental processes alone? For instance, I am not too convinced by the social aspect of courtship learning. It rather seems that repeated interactions with the same individual (which can

be considered a requirement for the definition of sociality) suppress the development of the duet dance rather than increase it (Table 3). That there are more individual dances with familiar partners is interesting, but then we must be careful about how we discuss these results. Is this 'social' learning supposed to be for the development of the duet dance, which is claimed to be the important aspect that leads to increased mating success? Or is it social facilitation of opportunity for practice (which is not the same thing as learning e.g. from a model). Certainly the term 'lessons' in the title suggests one individual knows something about the dance and is teaching it to the other, but the fact that more dances occur with familiar partners is not necessarily evidence of this.

Aside from these conceptual points, the experiments are well controlled and statistical methods appropriate. With some changes to the structure and argumentation I suggest this paper this can be accepted after changes.

Minor points.

"However, this phenomenon might be explained by the fact that we lacked data on very young birds in the paired condition (see methods)." This needs more explanation! Is there any way you could explore the impact of these missing data points on your statistical models?

As above, does family condition also contain potential duetting partners?

What is the potential effect of familiarity arising through auditory cues alone? Are the partners taken from within this connected auditory pool of animals?

Review form: Reviewer 3

Is the manuscript scientifically sound in its present form?

Yes

Are the interpretations and conclusions justified by the results?

No

Is the language acceptable?

Yes

Is it clear how to access all supporting data?

Yes

Do you have any ethical concerns with this paper?

No

Have you any concerns about statistical analyses in this paper?

Yes

Recommendation?

Major revision is needed (please make suggestions in comments)

Comments to the Author(s)

This study aims to test for age-related and social factors influence on dance development in Java sparrows. The idea of the manuscript is very interesting and scarce in current literature, therefore

I think the data should catch the attention of a wide audience. The authors provide a good introduction which clearly justify their work, however I had several concerns regarding their methods and, more specifically, their statistical analyses. I recommend they revise these points and provide more details about their approach. I made several specific comments below.

Title: "Early life" should be "early-life".

L 23: "but their developmental mechanisms have been identified only for vocalizations". I agree that this kind of information is lacking in literature, but later in this paragraph you've cited Hoepfner & Goller (2013) in which body motions were also investigated. Please, revise.

L 37, 39: Citations could be included in the statements below, since there is evidence for these ideas for other communication channels, such as the vocalization, and which are not necessarily related to "animal play".

- "communicative functions and is performed in front of conspecifics for socialization and future pair formation"

- "it is also possible that young birds need to 'practice' dance for motor training, just as they do for the acquisition of songs"

L 56: Replace "(Lonchura oryzivora; order: Passeriformes, family: Estrildidae)" by "(Lonchura oryzivora; Passeriformes: Estrildidae)"

L 69: After reading the whole manuscript I think you did not test for "timing" changes in dance practicing. If you intended to refer to "timing" as "age-related" changes, this is already mentioned in this sentence.

L 87: "When the subject birds were young": What do you mean by "young" here? How many dph?

L 103: What is one bout of dance? More than one dance sequence (as described in lines 98 and 99) or just one? Please, clarify.

L 106: This means that the previous GLMM included "number of dance bouts without songs"? Also, later in the results section you use "song-dance cooccurrence" to refer to dance bouts that accompanied song, please standardize the terms throughout the text to facilitate reading.

L 107: In L 87 you mentioned that "When the subject birds were young, we used the family condition," therefore, I believe that your treatments in "age" and "experimental condition" should be the same (i.e., correlated). Did you test for this? How many replicates in each treatment?

L 111: "repeated encounter with the same partner (order of pairing with the same partner)". I did not understand this variable, is this the number of times an individual met a specific partner or the ordinal position in which he repeatedly met a specific partner? Please, clarify. How was your experiment designed so that one male would repeatedly met another male? In lines 94-95 you mentioned that "Overall, each focal male was paired with each partner 1-19 times throughout the study." But what was the mean number of repeated partners for all focal males?

L 116-117: Your model looks overfitted, how many samples do you have? You've included five variables, but you have only nine focal males, this should be a problem and your statistical results may not be robust. I was not clear about your sample size throughout the manuscript.

L 120: Please provide more details on your statistical approach. How did you test for model

validity, overdispersion in Poisson models and which R packages did you use? How did you test for variables significance in models? There are several ways to do it using R. Please provide references for statistical analyses. I have several concerns regarding these analyses, see also my comments about Figure 2 (below).

L 140: The discussion needs to be revised after the appropriate statistical analyses, because results may change.

L 277: Table 1: Estimates for categorical variables are usually presents relative to one reference level. Here you presented "Condition (pair or family)", which one is the reference level?

Tables 1 and 2: What is your total sample size in all models?

L 310: Y-axis in binomial distribution GLMM (c and d) is easier to understand if presented as "probability of song-dance cooccurrence or duet dance occurrence". This is what the line refers to. Also, provide the y-axis scale (not only extreme values, 0 or 1).

Figure 2b. It looks like your data is "zero-inflated", therefore your GLMM should not be valid in its current presentation. You will probably need to use zero-inflated models in which a binomial and zero-truncated Poisson model are built. See Zuur et al (2009) "Mixed effects models and extensions in Ecology with R" for more details.

Figures 2b, c and d: You should present confidence intervals for all lines estimated from models and the variance of the random factor (bird identity).

Decision letter (RSOS-180980.R0)

04-Oct-2018

Dear Dr Soma:

Manuscript ID RSOS-180980 entitled "Early life lessons of the courtship dance in a dance-duetting songbird, the Java sparrow" which you submitted to Royal Society Open Science, has been reviewed. The comments from reviewers are included at the bottom of this letter.

In view of the criticisms of the reviewers, the manuscript has been rejected in its current form. However, a new manuscript may be submitted which takes into consideration these comments.

Please note that resubmitting your manuscript does not guarantee eventual acceptance, and that your resubmission will be subject to peer review before a decision is made.

Your resubmitted manuscript should be submitted by 03-Apr-2019. If you are unable to submit by this date please contact the Editorial Office.

Please note that Royal Society Open Science will introduce article processing charges for all new

submissions received from 1 January 2018. Charges will also apply to papers transferred to Royal Society Open Science from other Royal Society Publishing journals, as well as papers submitted as part of our collaboration with the Royal Society of Chemistry (<http://rsos.royalsocietypublishing.org/chemistry>). If your manuscript is submitted and accepted for publication after 1 Jan 2018, you will be asked to pay the article processing charge, unless you request a waiver and this is approved by Royal Society Publishing. You can find out more about the charges at <http://rsos.royalsocietypublishing.org/page/charges>. Should you have any queries, please contact openscience@royalsociety.org.

on behalf of Dr Kristina Sefc (Associate Editor) and Professor Kevin Padian (Subject Editor)
openscience@royalsociety.org

Subject Editor Comments:

Here I'm going to opt for a "reject/resub" decision because two of your referees have different view of your statistical analyses, and there are varied concerns of all three. The R/R decision gives you more time than the "major revision" one, and this may be more helpful. Please make sure to address all the comments of the reviewers. Thanks.

Associate Editor Comments to Author (Dr Kristina Sefc):

Associate Editor: 1

Comments to the Author:

The manuscript has been seen by three reviewers, who recommended it based on the interesting research question but raised major concerns regarding the statistical methods and the fit of the experimental design to the research question. It appears that some of these issues cannot be resolved without further information, e.g. regarding sample size, possible overfitting of models, exact specification of what was tested (for instance, there was uncertainty about the terms "timing", "occurrence", "onset"). In your revision, please provide the necessary information and clarifications and consider the suggested revision of the statistical analysis.

Reviewers' Comments to Author:

Reviewer: 1

Comments to the Author(s)

The manuscript reports on an observational study of dance development in the Java sparrow. Dance development in songbirds has been too little studied, and is potentially important especially because of possible comparisons with song development. A limitation of the present study is that only the simplest measurements of dance were made: whether dancing occurs, whether dancing is paired with song, and whether dancing occurs in a duet with a partner. Without more sophisticated measures of dance performance, the study cannot really show that performance improves through the period of practice. By contrast, studies of song duets have tested whether temporal coordination and duet code adherence improve over time. With dance duets, some analog of temporal coordination presumably could be measured, but that was not attempted here. A further step would be to study dance development experimentally, in the way

that has been with song development starting with the classic studies of Thorpe and Marler. For example, one could isolate young birds and study dance development in the absence of external models – but one would still have to have more sophisticated measures of the dance to decide whether the form of the dance is altered by isolation.

The authors do provide convincing evidence of certain developmental trends: the frequency of dancing goes up with age, co-occurrence of dancing with song increases with age, and duet dancing increases with age. But given the limitations of the dance measurements that are made, not a lot can be concluded from these results. There is very little indication on whether dancing skill increases with age, on whether practice is necessary to dance properly, or on whether young birds learn anything from observing others dance. The authors have no evidence on whether dance in young birds has a communicative function or is necessary for coordinated singing and dancing (hypotheses mentioned in their introduction). The developmental trends that are observed could be due simply to changes in motivation to dance combined with the known developmental patterns of song.

Lines 15-16: I don't see what in the results suggests that early social experience is essential for the development of courtship. The authors would have to make an explicit argument on this point if they want to include this claim.

Lines 36-40: these statements are hypotheses, not predictions. I'm not sure why these hypotheses are mentioned here since they are not tested in the paper.

Lines 58-59: the previous study does not show that duet dancing is crucial to mating success in the sense of causing higher mating success. The authors might say that duet dancing is associated with mating success in that study, but the results of that study do not show that duet dancing is causal.

Line 91: to say that birds were “audibly isolated” means that one could hear that they were isolated. A better wording would be to say is that the birds were not “acoustically isolated.”

Line 143: Can you give an estimate of time investment in dance practice? It does seem that you have the data to do so.

W. A. Searcy
University of Miami

Reviewer: 2

Comments to the Author(s)

This is a well written piece that touches on numerous aspects of important theory with respect to the development and expression of courtship behaviour in birds. I find the introduction to be very compelling, setting up the study nicely. The authors ask whether the 'dancing' aspects of courtship develop over time, in a similar fashion to courtship song, showing convincingly that it does. I am not a bird person so am not certain of their claims this is the first time this has been shown, but nevertheless the data support the claim. Secondly the authors ask whether social conditions affect the development of dance and the co-occurrence of song and dance. Here is where the study and interpretation get a bit more tricky. First, the authors change the housing conditions of juveniles for brief periods during development - keeping them with the family unit (which is confusingly not defined in the main text, does it contain other juvenile birds?) or the paired condition in which birds are kept with other juveniles (male or female). They ask whether

the onset of dance (confusingly called ‘occurrence’ but should probably be changed to ‘onset’), the rate of dance (‘number of dance bouts’), and the co-occurrence of song and dance are affected by these social contexts. From these treatments and measurements, my take away message is that the number of dances increases in the paired condition, which may be a consequence of greater opportunities or motivation when meeting a juvenile (because whether this is the only time I see a juvenile or not I cannot tell as I don’t understand whether the family condition also contains juveniles), or may be a function of the familiarity of the juvenile partner as the authors claim - the frequency of dancing may be higher when the focal bird does not know its partner, at least initially, but then becomes familiar and increases the frequency of bouts with better known birds. The co-occurrence of dance and song increases with age, but not with social conditions.

While I have some reservations about how well the methodology addresses some of the points raised in the introduction, I nevertheless think the paper warrants publication after some small changes to better explain the differences in conditions and opportunity for courtship interactions in the treatments. The main point I would like to see the authors address is how much evidence they have that there is any aspect of social learning going on that cannot be explained by developmental processes alone? For instance, I am not too convinced by the social aspect of courtship learning. It rather seems that repeated interactions with the same individual (which can be considered a requirement for the definition of sociality) suppress the development of the duet dance rather than increase it (Table 3). That there are more individual dances with familiar partners is interesting, but then we must be careful about how we discuss these results. Is this ‘social’ learning supposed to be for the development of the duet dance, which is claimed to be the important aspect that leads to increased mating success? Or is it social facilitation of opportunity for practice (which is not the same thing as learning e.g. from a model). Certainly the term ‘lessons’ in the title suggests one individual knows something about the dance and is teaching it to the other, but the fact that more dances occur with familiar partners is not necessarily evidence of this.

Aside from these conceptual points, the experiments are well controlled and statistical methods appropriate. With some changes to the structure and argumentation I suggest this paper this can be accepted after changes.

Minor points.

“However, this phenomenon might be explained by the fact that we lacked data on very young birds in the paired condition (see methods).” This needs more explanation! Is there any way you could explore the impact of these missing data points on your statistical models?

As above, does family condition also contain potential duetting partners?

What is the potential effect of familiarity arising through auditory cues alone? Are the partners taken from within this connected auditory pool of animals?

Reviewer: 3

Comments to the Author(s)

This study aims to test for age-related and social factors influence on dance development in Java sparrows. The idea of the manuscript is very interesting and scarce in current literature, therefore I think the data should catch the attention of a wide audience. The authors provide a good introduction which clearly justify their work, however I had several concerns regarding their methods and, more specifically, their statistical analyses. I recommend they revise these points and provide more details about their approach. I made several specific comments below.

Title: "Early life" should be "early-life".

L 23: "but their developmental mechanisms have been identified only for vocalizations". I agree that this kind of information is lacking in literature, but later in this paragraph you've cited Hoepfner & Goller (2013) in which body motions were also investigated. Please, revise.

L 37, 39: Citations could be included in the statements below, since there is evidence for these ideas for other communication channels, such as the vocalization, and which are not necessarily related to "animal play".

- "communicative functions and is performed in front of conspecifics for socialization and future pair formation"

- "it is also possible that young birds need to 'practice' dance for motor training, just as they do for the acquisition of songs"

L 56: Replace "(*Lonchura oryzivora*; order: Passeriformes, family: Estrildidae)" by "(*Lonchura oryzivora*; Passeriformes: Estrildidae)"

L 69: After reading the whole manuscript I think you did not test for "timing" changes in dance practicing. If you intended to refer to "timing" as "age-related" changes, this is already mentioned in this sentence.

L 87: "When the subject birds were young": What do you mean by "young" here? How many dph?

L 103: What is one bout of dance? More than one dance sequence (as described in lines 98 and 99) or just one? Please, clarify.

L 106: This means that the previous GLMM included "number of dance bouts without songs"? Also, later in the results section you use "song-dance cooccurrence" to refer to dance bouts that accompanied song, please standardize the terms throughout the text to facilitate reading.

L 107: In L 87 you mentioned that "When the subject birds were young, we used the family condition", therefore, I believe that your treatments in "age" and "experimental condition" should be the same (i.e., correlated). Did you test for this? How many replicates in each treatment?

L 111: "repeated encounter with the same partner (order of pairing with the same partner)". I did not understand this variable, is this the number of times an individual met a specific partner or the ordinal position in which he repeatedly met a specific partner? Please, clarify. How was your experiment designed so that one male would repeatedly met another male? In lines 94-95 you mentioned that "Overall, each focal male was paired with each partner 1-19 times throughout the study." But what was the mean number of repeated partners for all focal males?

L 116-117: Your model looks overfitted, how many samples do you have? You've included five variables, but you have only nine focal males, this should be a problem and you statistical results may not be robust. I was not clear about your sample size throughout the manuscript.

L 120: Please provide more details on your statistical approach. How did you test for model validity, overdispersion in Poisson models and which R packages did you use? How did you test for variables significance in models? There are several ways to do it using R. Please provide references for statistical analyses. I have several concerns regarding these analyses, see also my comments about Figure 2 (below).

L 140: The discussion needs to be revised after the appropriate statistical analyses, because results may change.

L 277: Table 1: Estimates for categorical variables are usually presents relative to one reference level. Here you presented "Condition (pair or family)", which one is the reference level?

Tables 1 and 2: What is your total sample size in all models?

L 310: Y-axis in binomial distribution GLMM (c and d) is easier to understand if presented as "probability of song-dance cooccurrence or duet dance occurrence". This is what the line refers to. Also, provide the y-axis scale (not only extreme values, 0 or 1).

Figure 2b. It looks like your data is "zero-inflated", therefore your GLMM should not be valid in its current presentation. You will probably need to use zero-inflated models in which a binomial and zero-truncated Poisson model are built. See Zuur et al (2009) "Mixed effects models and extensions in Ecology with R" for more details.

Figures 2b, c and d: You should present confidence intervals for all lines estimated from models and the variance of the random factor (bird identity).

Author's Response to Decision Letter for (RSOS-180980.R0)

See Appendix A.

RSOS-190563.R0

Review form: Reviewer 1 (William Searcy)

Is the manuscript scientifically sound in its present form?

Yes

Are the interpretations and conclusions justified by the results?

Yes

Is the language acceptable?

Yes

Is it clear how to access all supporting data?

Yes

Do you have any ethical concerns with this paper?

No

Have you any concerns about statistical analyses in this paper?

No

Recommendation?

Accept with minor revision (please list in comments)

Comments to the Author(s)

The authors have done an excellent job of addressing my criticisms of the previous version of their manuscript. In particular, the addition of an analysis of dance speed and dance duration answers my objection that dancing skill had not been measured; in my opinion this addition adds substantially to the value of the paper.

At this stage I have only some minor suggestions on wording.

Line 10: perhaps it would make more sense to say "we propose that the practice is needed for motor learning."

Line 21: I find "courtship/mating dance/display" confusing. I suggest simplifying this phrase.

Line 26: I suggest "By contrast" rather than "Conversely."

Line 38: "they" here has no obvious referent. I suggest substituting "practice dances" if that is what you intend.

Line 41: "become" not "becomes"

Line 51: "the duets of adults" rather than "adults duets" – though you could use "adults' duets"

Line 87: do you mean "temporary isolation" rather than "temporal isolation"?

Line 92: "raised in a different family"

Line 93: "isolated from others" rather than "isolated by others"

Lines 119-120: I am unsure of the meaning of "ordinal position in which he repeatedly met a specific partner." Can you clarify?

Line 155: perhaps you mean "elicited" rather than "solicited"

Line 175: I suggest "call" rather than "called"

Line 192: I suggest "it also remains unknown why..."

Lines 203-204: "is improvised rather than stereotyped"

Review form: Reviewer 3

Is the manuscript scientifically sound in its present form?

Yes

Are the interpretations and conclusions justified by the results?

Yes

Is the language acceptable?

Yes

Is it clear how to access all supporting data?

Yes

Do you have any ethical concerns with this paper?

No

Have you any concerns about statistical analyses in this paper?

Yes

Recommendation?

Accept with minor revision (please list in comments)

Comments to the Author(s)

This is my second review of this manuscript and I consider that authors did a very good job by taking into account almost all comments provided by reviewers. My questions were almost completely answered and I recognize author's effort when dealing with their analyses, which required a more advanced method giving the data nature.

Although authors were not able to use a Zero-inflated model, because of their sample size and techniques implemented in statistical software, I was almost convinced that the zero-inflated approach is not necessary. However, I was curious if their current models were validated, because 95% Confidence Intervals in Figures 2b and 2c look wrong to me. They are too wide and, if that are the correct CIs, then I think the models have serious problems and do not reflect the best analyses for this data. Please check these CI and provide convincing information that these models are validated. When fitting the CI, be sure you are using the correct link function of the model (and please provide information about which one is this link function). The CI should be relative to model fitted values and not the raw data.

Also, given the fact that sample size (number of different individuals) is small, I would leave this information even more clear in the text, so authors may conclude by themselves about data quality. Therefore, I recommend you specify the number of sampled birds in Tables 1, 2 and 4 similarly as you did in Table 3.

Two minor comments:

L 329 Indicates what terms in bold mean (=reference level).

Table 1: briefly explain in the legend what is "subset data".

Decision letter (RSOS-190563.R0)

07-May-2019

Dear Dr Soma

On behalf of the Editor, I am pleased to inform you that your Manuscript RSOS-190563 entitled "Early-life lessons of the courtship dance in a dance-duetting songbird, the Java sparrow" has been accepted for publication in Royal Society Open Science subject to minor revision in accordance with the referee suggestions. Please find the referees' comments at the end of this email.

The reviewers and Subject Editor have recommended publication, but also suggest some minor revisions to your manuscript. Therefore, I invite you to respond to the comments and revise your manuscript.

- Ethics statement

- Data accessibility

It is a condition of publication that all supporting data are made available either as supplementary information or preferably in a suitable permanent repository. The data

accessibility section should state where the article's supporting data can be accessed. This section should also include details, where possible of where to access other relevant research materials such as statistical tools, protocols, software etc can be accessed. If the data has been deposited in an external repository this section should list the database, accession number and link to the DOI for all data from the article that has been made publicly available. Data sets that have been deposited in an external repository and have a DOI should also be appropriately cited in the manuscript and included in the reference list.

If you wish to submit your supporting data or code to Dryad (<http://datadryad.org/>), or modify your current submission to dryad, please use the following link:
<http://datadryad.org/submit?journalID=RSOS&manu=RSOS-190563>

- **Competing interests**

- **Authors' contributions**

- **Acknowledgements**

- **Funding statement**

Because the schedule for publication is very tight, it is a condition of publication that you submit the revised version of your manuscript before 16-May-2019. Please note that the revision deadline will expire at 00.00am on this date. If you do not think you will be able to meet this date please let me know immediately.

To revise your manuscript, log into <https://mc.manuscriptcentral.com/rsos> and enter your Author Centre, where you will find your manuscript title listed under "Manuscripts with Decisions". Under "Actions," click on "Create a Revision." You will be unable to make your

revisions on the originally submitted version of the manuscript. Instead, revise your manuscript and upload a new version through your Author Centre.

on behalf of Dr Kristina Sefc (Associate Editor) and Kevin Padian (Subject Editor)
openscience@royalsociety.org

Associate Editor Comments to Author (Dr Kristina Sefc):

Dear authors,
I am happy to inform you that both reviewers agree with your revision, and ask you to take care of their remaining concerns and suggestions before you resubmit your manuscript.
Best regards, Kristina

Reviewer comments to Author:

Reviewer: 1

Comments to the Author(s)

The authors have done an excellent job of addressing my criticisms of the previous version of their manuscript. In particular, the addition of an analysis of dance speed and dance duration answers my objection that dancing skill had not been measured; in my opinion this addition adds substantially to the value of the paper.

At this stage I have only some minor suggestions on wording.

Line 10: perhaps it would make more sense to say "we propose that the practice is needed for motor learning."

Line 21: I find "courtship/mating dance/display" confusing. I suggest simplifying this phrase.

Line 26: I suggest "By contrast" rather than "Conversely."

Line 38: "they" here has no obvious referent. I suggest substituting "practice dances" if that is what you intend.

Line 41: "become" not "becomes"

Line 51: "the duets of adults" rather than "adults duets" – though you could use "adults' duets"

Line 87: do you mean "temporary isolation" rather than "temporal isolation"?

Line 92: "raised in a different family"

Line 93: "isolated from others" rather than "isolated by others"

Lines 119-120: I am unsure of the meaning of "ordinal position in which he repeatedly met a specific partner." Can you clarify?

Line 155: perhaps you mean "elicited" rather than "solicited"

Line 175: I suggest "call" rather than "called"

Line 192: I suggest "it also remains unknown why..."

Lines 203-204: "is improvised rather than stereotyped"

Reviewer: 3

Comments to the Author(s)

This is my second review of this manuscript and I consider that authors did a very good job by taking into account almost all comments provided by reviewers. My questions were almost completely answered and I recognize author's effort when dealing with their analyses, which required a more advanced method giving the data nature.

Although authors were not able to use a Zero-inflated model, because of their sample size and techniques implemented in statistical software, I was almost convinced that the zero-inflated approach is not necessary. However, I was curious if their current models were validated, because 95% Confidence Intervals in Figures 2b and 2c look wrong to me. They are too wide and, if that are the correct CIs, then I think the models have serious problems and do not reflect the best analyses for this data. Please check these CI and provide convincing information that these models are validated. When fitting the CI, be sure you are using the correct link function of the model (and please provide information about which one is this link function). The CI should be relative to model fitted values and not the raw data.

Also, given the fact that sample size (number of different individuals) is small, I would leave this information even more clear in the text, so authors may conclude by themselves about data quality. Therefore, I recommend you specify the number of sampled birds in Tables 1, 2 and 4 similarly as you did in Table 3.

Two minor comments:

L 329 Indicates what terms in bold mean (=reference level).

Table 1: briefly explain in the legend what is "subset data".

Author's Response to Decision Letter for (RSOS-190563.R0)

See Appendix B.

Decision letter (RSOS-190563.R1)

13-May-2019

Dear Dr Soma,

I am pleased to inform you that your manuscript entitled "Early-life lessons of the courtship dance in a dance-duetting songbird, the Java sparrow" is now accepted for publication in Royal Society Open Science.

on behalf of Dr Kristina Sefc (Associate Editor) and Kevin Padian (Subject Editor)
openscience@royalsociety.org

Appendix A

Department of Biology Faculty of Science

N10 W8, Kita-ku, Sapporo, Japan 060-0810

Tel: +81-(0)-11-706-2995

www.hokudai.ac.jp

March 26th, 2019

Dear Editor,

We are delighted to be given an opportunity to revise and resubmit the manuscript.

We are really grateful to the three reviewers and to you for recognizing the potential significance of the outcomes of this work and providing us with very constructive comments and suggestions. We have revised the manuscript and added analyses, following the comments of the reviewers. All revisions in response to the reviewers' comments are explained in the end of this letter, and the corresponding parts in the manuscript are colored. We hope the manuscript will be accepted for publication in Royal Society Open Science.

Sincerely yours,

Masayo Soma

Response to associate editor comment (Dr Kristina Sefc):

The manuscript has been seen by three reviewers, who recommended it based on the interesting research question but raised major concerns regarding the statistical methods and the fit of the experimental design to the research question. It appears that some of these issues cannot be resolved without further information, e.g. regarding sample size, possible overfitting of models, exact specification of what was tested (for instance, there was uncertainty about the terms “timing”, “occurrence”, “onset”). In your revision, please provide the necessary information and clarifications and consider the suggested revision of the statistical analysis.

→ We carefully revised the manuscript in response to all the reviewers’ concerns. Thanks to their inputs, we believe that the revised manuscript gained clarity and necessary evidence. Detailed revisions are explained below.

Response to reviewer 1 comments:

The manuscript reports on an observational study of dance development in the Java sparrow. Dance development in songbirds has been too little studied, and is potentially important especially because of possible comparisons with song development. A limitation of the present study is that only the simplest measurements of dance were made: whether dancing occurs, whether dancing is paired with song, and whether dancing occurs in a duet with a partner. Without more sophisticated measures of dance performance, the study cannot really show that performance improves through the period of practice. By contrast, studies of song duets have tested whether temporal coordination and duet code adherence improve over time. With dance duets, some analog of temporal coordination presumably could be measured, but that was not attempted here. A further step would be to study dance development experimentally, in the way that has been with song development starting with the classic studies of Thorpe and Marler. For example, one could isolate young birds and study dance development in the absence of external models – but one would still have to have more sophisticated measures of the dance to decide whether the form of the dance is altered by isolation.

The authors do provide convincing evidence of certain developmental trends: the frequency of dancing goes up with age, co-occurrence of dancing with song increases with age, and duet dancing increases with age. But given the limitations of the dance measurements that are made, not a lot can be concluded from these results. There is very little indication on whether dancing skill increases with age, on whether practice is necessary to dance properly, or on whether young birds learn anything from observing others dance. The authors have no evidence on whether dance in young birds has a communicative function or is necessary for coordinated singing and dancing (hypotheses mentioned in their introduction). The developmental trends that are observed could be due simply to changes in motivation to dance combined with the known developmental patterns of song.

→ It is a great honor for us to have a chance to get our work reviewed by Prof. Searcy. Thank you.

→ The reviewer’s main concerns raised here are sorted into three aspects and discussed as below.

[Dance measurements] We totally agree with the reviewer that dance performance should be measured precisely. We focused on three subject birds, for which we have abundant dance samples throughout the observation period, and examined age-related changes in dance speed (hop rate and bill-wipe rate) and dance bout duration. In accordance with our prediction that practicing dance would contribute to motor performance, we observed age-related increase in dance tempo and bout duration. We believe that this new finding would add scientific value. The manuscript was revised accordingly (L10-12, 133-140, 167-173, 183, Table 4, Figure 3).

[Temporal measurement of duetting] We admit that it is crucial to determine ‘duetting’ for better understanding of courtship communication in the Java sparrow. However, there are several methodological issues that cannot be easily solved. As mentioned in the manuscript, temporal

structure of dancing is not fixed, which is the main difference between dance of the Java sparrow and well-known examples of song duetting. We will definitely need a sophisticated modeling approach to describe temporal structures of dancing, for which we will need a large number of dance samples per individual. Only after establishing the methodology that can be applicable to adults, we will be able to look at developmental aspects. We are very careful because we would like to avoid type I error (erroneously detecting the presence of duet). The new results on dance performance revealed that the birds show higher dance speed, and longer dance duration as they get older. These changes can increase the chance that their dance motions apparently coincide with those of their partners, where it is difficult to tell whether they actually improved their ability to do ‘duet’. The present definition of duet is rather rough, but is robust under the situation described above.

[Social isolation] We are aware that more experimental approach, such as social isolation, would be needed for explicit test of experience-based development of courtship dancing, but unfortunately it is not feasible. Actually, we already had a chance to observe the behavior of the Java sparrows that were raised under social isolation in our colleague’s lab. Socially isolated Java sparrows show strong tendency of social phobia and behave extremely abnormally, where it is practically impossible to observe normal courtship interactions. Even if we managed to get some behavioral data from them, we could not tell if the poor courtship is merely due to the lack of performance skill or reflects psychological problems. Unlike songs that can be produced in solitary conditions, dancing always has to involve social interactions, which makes this problem difficult. For ethical reasons, currently we do not have a plan to have socially isolated Java sparrows. In the present manuscript, we explained that our study does have a limitation (L15-17), and needs experimental controls of social experience (L209-219). We expect that moderate manipulation, such as the control of the number of siblings or adults (parent), might work.

Lines 15-16: I don’t see what in the results suggests that early social experience is essential for the development of courtship. The authors would have to make an explicit argument on this point if they want to include this claim.

→ We agree that the sentence lacked evidence. The last part of the summary was revised and now says “our results do not show how much social experiences account for the development of dance communication” (L15-16).

Lines 36-40: these statements are hypotheses, not predictions. I’m not sure why these hypotheses are mentioned here since they are not tested in the paper.

→ [*1] Among the three hypotheses that we mentioned in the previous manuscript, the first one (possible role for communication) was deleted. As the reviewer said, the present study did not test this aspect. For the remaining two (song-dance coordination, motor performance), our

study presents relevant findings (figure 2b & newly added figure 3). These results are far from conclusive because they are descriptive and lack experimental manipulations, but at least we can say that the phenomena we observed were consistent with what we could expect from these ideas. Just like the classic (and descriptive) birdsong studies in early time that served as a foundation of the science of vocal learning, we hope and believe that this work could contribute to the future development of the research area.

→ The word ‘predictions’ was substituted with ‘hypotheses’.

Lines 58-59: the previous study does not show that duet dancing is crucial to mating success in the sense of causing higher mating success. The authors might say that duet dancing is associated with mating success in that study, but the results of that study do not show that duet dancing is causal.

→ We revised the misleading sentence (L57-58). Now it just says that duet dancing precedes successful mating. We appreciate the reviewer’s effort to read our previous paper for this review.

Line 91: to say that birds were “audibly isolated” means that one could hear that they were isolated. A better wording would be to say is that the birds were not “acoustically isolated.”

→ Thank you for the clarification. We revised the part as suggested (L94).

Line 143: Can you give an estimate of time investment in dance practice? It does seem that you have the data to do so.

→ As aforementioned, we conducted additional detailed analyses on dance, which enabled us to estimate the time investment. At peak, one male danced for 13.7 minutes (= 823.4 sec) in total during 5-hour observation. We mentioned it in the result section (L168-169). In addition, the fact that the Java sparrows start dancing early in life could also be seen as a time investment.

Response to reviewer 2 comments:

This is a well written piece that touches on numerous aspects of important theory with respect to the development and expression of courtship behaviour in birds. I find the introduction to be very compelling, setting up the study nicely. The authors ask whether the ‘dancing’ aspects of courtship develop over time, in a similar fashion to courtship song, showing convincingly that it does. I am not a bird person so am not certain of their claims this is the first time this has been shown, but nevertheless the data support the claim. Secondly the authors ask whether social conditions affect the development of dance and the co-occurrence of song and dance. Here is where the study and interpretation get a bit more tricky. First, the authors change the housing conditions of juveniles for brief periods during development - keeping them with the family unit (which is confusingly not defined in the main text, does it contain other juvenile birds?) or the paired condition in which birds are kept with other juveniles (male or female). They ask whether the onset of dance (confusingly called ‘occurrence’ but should probably be changed to ‘onset’), the rate of dance (‘number of dance bouts’), and the co-occurrence of song and dance are affected by these social contexts. From these treatments and measurements, my take away message is that the number of dances increases in the paired condition, which may be a consequence of greater opportunities or motivation when meeting a juvenile (because whether this is the only time I see a juvenile or not I cannot tell as I don’t understand whether the family condition also contains juveniles), or may be a function of the familiarity of the juvenile partner as the authors claim - the

frequency of dancing may be higher when the focal bird does not know its partner, at least initially, but then becomes familiar and increases the frequency of bouts with better known birds. The co-occurrence of dance and song increases with age, but not with social conditions.

- We really appreciate that the reviewer sees some scientific values in our study.
- We added information on the family members that the subject birds grew up with (L76-77). The subject birds were always with 1-2 siblings (brood size = 2-3) in addition to the parents throughout their developmental period except when they were temporarily separated for the observation under paired condition (L85-86). This means that the subjects did have abundant chances to dance with juveniles (and actually did so in their family cages, with less frequency than in the paired condition though), and that the paired condition was not their first time to see juvenile birds.
- We believe that the term ‘occurrence’ is relevant for describing the measurements that we took. To avoid confusion, we also added an explanation that it means presence/absence (L107-108)

While I have some reservations about how well the methodology addresses some of the points raised in the introduction, I nevertheless think the paper warrants publication after some small changes to better explain the differences in conditions and opportunity for courtship interactions in the treatments. The main point I would like to see the authors address is how much evidence they have that there is any aspect of social learning going on that cannot be explained by developmental processes alone? For instance, I am not too convinced by the social aspect of courtship learning. It rather seems that repeated interactions with the same individual (which can be considered a requirement for the definition of sociality) suppress the development of the duet dance rather than increase it (Table 3). That there are more individual dances with familiar partners is interesting, but then we must be careful about how we discuss these results. Is this ‘social’ learning supposed to be for the development of the duet dance, which is claimed to be the important aspect that leads to increased mating success? Or is it social facilitation of opportunity for practice (which is not the same thing as learning e.g. from a model). Certainly the term ‘lessons’ in the title suggests one individual knows something about the dance and is teaching it to the other, but the fact that more dances occur with familiar partners is not necessarily evidence of this.

Aside from these conceptual points, the experiments are well controlled and statistical methods appropriate. With some changes to the structure and argumentation I suggest this paper this can be accepted after changes.

- This issue is very important, but hard to be addressed from the framework of this study. It’s true that social facilitation alone could explain the phenomena we observed, but we suspect that there might be more.

Songbirds (including Java sparrows and their related species like zebra finches) practice singing when alone. They learn (memorize) songs from social stimuli, but do not necessarily need social interactions for sensory-motor training. They seem to prefer quiet environment without interrupting conspecifics for practice singing. In contrast, we hardly (almost never) observed Java sparrows’ practicing dancing when they are alone. It seems that they always need partners for dancing. If dance practice were just for motor training, Java sparrows would practice dancing alone just like they do so for singing.

- We know that this evidence is too weak, but explained it in the manuscript (L194-196). We also clearly stated that what is learned is yet to be determined (L15-16, 209-211).

“However, this phenomenon might be explained by the fact that we lacked data on very young birds in the paired condition (see methods).” This needs more explanation! Is there any way you could explore the impact of these missing data points on your statistical models?

→ Unfortunately, it is practically impossible to add data points under pair condition in young age, because of the reason described in the manuscript. However, we could add a supplemental analysis. For the four birds that constituted pilot study, we have data points under both family and paired conditions between 60-180 dph (L89-92). So, we extracted the data from the later period (i.e., 60-180dph) of these birds (L110-116), and ran the same analysis as Table 1a. As shown in Table 1b, the outcomes are quite consistent between Table 1a and b.

As above, does family condition also contain potential duetting partners?

→ Yes. As mentioned earlier, we specified that the subject birds grew up with their siblings (L76-77).

What is the potential effect of familiarity arising through auditory cues alone? Are the partners taken from within this connected auditory pool of animals?

→ All the birds in our study are kept in one room, where they share the same auditory environment, which is now explained in the manuscript (L77, 93). Though we cannot discuss anything about auditory-based individual recognition in this study, we speculate that it is not a simple story. In the song-learning period, juveniles of *Lonchura* species learn both songs and calls. Therefore, the acoustic structures of contact calls change dramatically even within individuals during the developmental period (cf. Yoneda & Okanoya 1991, J Ethol).

Response to reviewer 3 comments:

This study aims to test for age-related and social factors influence on dance development in Java sparrows. The idea of the manuscript is very interesting and scarce in current literature, therefore I think the data should catch the attention of a wide audience. The authors provide a good introduction which clearly justify their work, however I had several concerns regarding their methods and, more specifically, their statistical analyses. I recommend they revise these points and provide more details about their approach. I made several specific comments below.

→ We are grateful to the reviewer for giving us many useful suggestions especially on statistics. We reanalyzed the data in accordance with those suggestions, and revised the manuscript accordingly.

Title: “Early life” should be “early-life”.

→ Title was revised as suggested.

L 23: “but their developmental mechanisms have been identified only for vocalizations”. I agree that this kind of information is lacking in literature, but later in this paragraph you’ve cited Hoepfner & Goller (2013) in which body motions were also investigated. Please, revise.

→ We agree that the expression was overly strong in the previous manuscript. We revised it (L23).

L 37, 39: Citations could be included in the statements below, since there is evidence for these ideas for other communication channels, such as the vocalization, and which are not necessarily related to “animal play”.

- “communicative functions and is performed in front of conspecifics for socialization and future pair formation”
- “it is also possible that young birds need to ‘practice’ dance for motor training, just as they do for the acquisition of songs”

→ The paragraph was revised in response to the reviewer 1’s comment (please see [*1]), and does not include the sentence on communicative functions any more. We added citations for the second sentence (L38).

L 56: Replace “(Lonchura oryzivora; order: Passeriformes, family: Estrildidae)” by “(Lonchura oryzivora; Passeriformes: Estrildidae)”

→ Revised as suggested (L55).

L 69: After reading the whole manuscript I think you did not test for “timing” changes in dance practicing. If you intended to refer to “timing” as “age-related” changes, this is already mentioned in this sentence.

→ As suggested, “timing” was deleted.

L 87: “When the subject birds were young”: What do you mean by “young” here? How many dph?

→ We added age information (L92)

L 103: What is one bout of dance? More than one dance sequence (as described in lines 98 and 99) or just one? Please, clarify.

→ Yes, the definition of one dance bout is as is given in the previous section. We specified it by saying “One dance bout was defined as...”(L102).

→ However, as will be explained later, we removed the poisson-GLMM result, which was turned out to be seriously affected by overdispersion. The corresponding parts in the method section were changed accordingly (L107-108).

L 106: This means that the previous GLMM included “number of dance bouts without songs”? Also, later in the results section you use “song-dance cooccurrence” to refer to dance bouts that accompanied song, please standardize the terms throughout the text to facilitate reading.

→ We revised the part to use consistent terminology and careful description of the dependent variable (L114-116).

L 107: In L 87 you mentioned that “When the subject birds were young, we used the family condition,”, therefore, I believe that your treatments in “age” and “experimental condition” should be the same (i.e., correlated) . Did you test for this? How many replicates in each treatment?

→ We agree that the two explanatory variables (age and pair/family condition) are confounding.

To deal with this, we repeated the same analysis on the subset data (please see L111-), because pilot study birds were observed under both paired and family conditions between 60–180 dph (L89-92). Table 1a and b show that the results are consistent, and so we think that our conclusion was not biased by undesirable set of predictor variables.

L 111: “repeated encounter with the same partner (order of pairing with the same partner)”. I did not understand this variable, is this the number of times an individual met a specific partner or the ordinal position in which he repeatedly met a specific partner? Please, clarify. How was your experiment designed so that one male would repeatedly met another male? In lines 94-95 you mentioned that “Overall, each focal male was paired with each partner 1–19 times throughout the study.” But what was the mean number of repeated partners for all focal males?

- This variable means the latter. To clarify it, we borrowed the reviewer’s expression (L120-121).
- We also showed the average number of the times that each pair met (L97-98).

L 116-117: Your model looks overfitted, how many samples do you have? You’ve included five variables, but you have only nine focal males, this should be a problem and you statistical results may not be robust. I was not clear about your sample size throughout the manuscript.

- This analysis is based on 46 data points taken from 6 subjects (the other 3 did not dance in the paired condition), which is now explained in the manuscript. We understand that such analysis would generate concern about overfitting. To deal with it, we repeated the analyses using simpler models. Specifically, we built four models, each containing age and one explanatory variable, and confirmed that the outcomes were consistent (L131-132). Age was always included as an explanatory variable in the above models because we would like to clarify whether increase in duetting is dependent on age, or some other explanatory factors (i.e. number of dance, song) that vary with age.
- In addition, we also checked for multicollinearity by calculating VIF (Variance Inflation Factor). VIFs were all < 5 , showing no sign of multicollinearity (L129-130).

L 120: Please provide more details on your statistical approach. How did you test for model validity, overdispersion in Poisson models and which R packages did you use? How did you test for variables significance in models? There are several ways to do it using R. Please provide references for statistical analyses. I have several concerns regarding these analyses, see also my comments about Figure 2 (below).

- Following the advice, we ran overdispersion test for each model. Overdispersion was detected for only count-data analyses (poisson GLMM). As this seems to be caused by zero-inflated data, and extremely high values for a few data points, we took the following two approaches
 - (1) We used binary data instead of counts (e.g. presence/absence of dance instead of dance counts), which was effective because such binomial GLMM showed no severe overdispersion at least with our data.
 - (2) We also tried several zero-inflated poisson (ZIP) models, which is explained in detail as a response to the later comment (please see [*2] below).

- As ZIP modeling had some limitations and was not ideal, we decided to adopt (1) to be shown in the manuscript. Therefore Table 1b and Fig 2b in the previous manuscript were deleted, and Table 2 shows binomial model instead of poisson.
- All necessary information, such as R package, references, and statistical significance, is now added in the method section (L106, 139,142-146, reference #32-35).

L 140: The discussion needs to be revised after the appropriate statistical analyses, because results may change.

- As explained above and below, main findings did not change after we had carefully reconsidered statistical procedures and made some modifications on them.

L 277: Table 1: Estimates for categorical variables are usually presents relative to one reference level. Here you presented “Condition (pair or family)”, which one is the reference level?

- It is explained in the table legend: “Positive values for the coefficient of condition indicate increases in paired condition.” We also used the bold fonts for ‘pair’ to stress it in the present manuscript.

Tables 1 and 2: What is your total sample size in all models?

- We added sample size for each table.

L 310: Y-axis in binomial distribution GLMM (c and d) is easier to understand if presented as “probability of song-dance cooccurrence or duet dance occurrence”. This is what the line refers to. Also, provide the y-axis scale (not only extreme values, 0 or 1).

- Y axis labels and scales were changed exactly as suggested.

Figure 2b. It looks like your data is “zero-inflated”, therefore your GLMM should not be valid in its current presentation. You will probably need to use zero-inflated models in which a binomial and zero-truncated Poisson model are built. See Zuur et al (2009) “Mixed effects models and extensions in Ecology with R” for more details.

- [*2] Yes, the data contain many zeros, which is because some subject birds hardly showed dance during the observation. Following the advice, we struggled with several zero-inflated poisson (ZIP) models described in Zuur et al. (2009) and elsewhere. The problem with the model in Zuur et al. (2009) (package ‘pscl’) is that it does not allow the entry of random effects (controlling for bird ID effect in our case). We searched for packages that can handle ZIP and random effects, and found ‘GLMMadaptive’. However, this package frequently returned errors seemingly due to the difficulty in model convergence, and we could not enter two explanatory variables at the same time. So, we had to build models as follows:

- ‘pscl’ model [fixed effect: age, cage condition, ID]
- ‘GLMMadaptive’ model [fixed effect: age or cage condition, random effect: ID]

As shown below, ZIP model outcomes fit with the result of binomial GLMM (Table 1a).

ZIP(psci)

Count model coefficients

	Estimate	SE	z	p
(Intercept)	-0.314	0.390	-0.806	0.420
age	0.003	0.002	1.307	0.191
Condition	1.641	0.366	4.479	<0.001*

Zero-inflation model coefficients (ID)

	Estimate	SE	z	p
(Intercept)	0.049	0.408	0.120	0.904
IDJS0295	0.337	0.599	0.562	0.574
IDJS0299	1.478	0.684	2.162	0.031*
IDJS0300	1.377	0.607	2.270	0.023*
IDJS0313	18.161	2353.657	0.008	0.994
IDJS0317	-0.300	0.655	-0.459	0.646
IDJS0318	18.161	2110.565	0.009	0.993
IDJS0320	-0.191	0.697	-0.274	0.784
IDJS0324	0.440	0.661	0.665	0.506

ZIP(GLMMadaptive)

Age effect (random effect: ID)

	Estimate	SE	z	p
(Intercept)	-0.624	0.243	-2.563	0.010
age	0.008	0.002	3.867	<0.001*

Condition effect (random effect: ID)

	Estimate	Std.Err	z-value	p-value
(Intercept)	-0.730	0.389	-1.877	0.061
Condition	1.813	0.398	4.557	<0.001*

Figures 2b, c and d: You should present confidence intervals for all lines estimated from models and the variance of the random factor (bird identity).

→ Following the suggestion, we added 95% CI lines for all the figures, and random factor variance for all tables.

Appendix B

Department of Biology Faculty of Science

N10 W8, Kita-ku, Sapporo, Japan 060-0810

Tel: +81-(0)-11-706-2995

www.hokudai.ac.jp

May 10th, 2019

Dear Editor,

We feel so much delighted to hear that our manuscript has been accepted for publication.

We are really grateful to the reviewers and to you for recognizing the scientific significance of this work. We have revised the manuscript in accordance with the suggestions of the reviewers. All revisions in response to the reviewers' comments are explained in the end of this letter, and the corresponding parts in the manuscript are colored.

Sincerely yours,

Masayo Soma

Response to reviewer 1 comments:

The authors have done an excellent job of addressing my criticisms of the previous version of their manuscript. In particular, the addition of an analysis of dance speed and dance duration answers my objection that dancing skill had not been measured; in my opinion this addition adds substantially to the value of the paper. At this stage I have only some minor suggestions on wording.

Line 10: perhaps it would make more sense to say “we propose that the practice is needed for motor learning.”

Line 21: I find “courtship/mating dance/display” confusing. I suggest simplifying this phrase.

Line 26: I suggest “By contrast” rather than “Conversely.”

Line 38: “they” here has no obvious referent. I suggest substituting “practice dances” if that is what you intend.

Line 41: “become” not “becomes”

Line 51: “the duets of adults” rather than “adults duets” – though you could use “adults’ duets”

Line 87: do you mean “temporary isolation” rather than “temporal isolation”?

Line 92: “raised in a different family”

Line 93: “isolated from others” rather than “isolated by others”

Lines 119-120: I am unsure of the meaning of “ordinal position in which he repeatedly met a specific partner.” Can you clarify?

Line 155: perhaps you mean “elicited” rather than “solicited”

Line 175: I suggest “call” rather than “called”

Line 192: I suggest “it also remains unknown why...”

Lines 203-204: “is improvised rather than stereotyped”

→ We are really grateful to the reviewer for giving us detailed suggestions. Every part was revised exactly as suggested.

Response to reviewer 3 comments:

This is my second review of this manuscript and I consider that authors did a very good job by taking into account almost all comments provided by reviewers. My questions were almost completely answered and I recognize author’s effort when dealing with their analyses, which required a more advanced method giving the data nature.

Although authors were not able to use a Zero-inflated model, because of their sample size and techniques implemented in statistical software, I was almost convinced that the zero-inflated approach is not necessary. However, I was curious if their current models were validated, because 95% Confidence Intervals in Figures 2b and 2c look wrong to me. They are too wide and, if that are the correct CIs, then I think the models have serious problems and do not reflect the best analyses for this data. Please check these CI and provide convincing information that these models are validated. When fitting the CI, be sure you are using the correct link function of the model (and please provide information about which one is this link function). The CI should be relative to model fitted values and not the raw data.

→ We may have used a wrong method for calculating CI, as CI estimation for mixed models is complicated and require advanced techniques. Now we plotted them based on GLM as shown in the manuscript.

Also, given the fact that sample size (number of different individuals) is small, I would leave this information even more clear in the text, so authors may conclude by themselves about data quality. Therefore, I recommend you specify the number of sampled birds in Tables 1, 2 and 4 similarly as you did in Table 3.

→ Required sample size (number of birds) is added.

L 329 Indicates what terms in bold mean (=reference level).

Table 1: briefly explain in the legend what is “subset data”.

→ Revised following the advices (L323-324)